# Monitoring the temperature dependent elastic and anelastic properties in isotropic polycrystalline ice using resonant ultrasound spectroscopy

Matthew J. Vaughan[1], Kasper van Wijk[2], David J. Prior[1], and M. Hamish Bowman[1]

[1]Department of Geology, University of Otago, 360 Leith Walk, Dunedin, New Zealand, 9054.
[2]Department of Physics, Building 303, University of Auckland, 38 Princes Street, Auckland, New Zealand, 92019
*Correspondence to:* Matthew Vaughan (mattvaughan902@gmail.com)

**Abstract.**

The elastic and anelastic properties of ice are of interest in the study of the dynamics of sea ice, glaciers and ice sheets. Resonant ultrasound spectroscopy allows quantitative estimates of these properties and aids calibration of active and passive seismic data gathered in the field. The elastic constants and the seismic quality factor $Q$ in laboratory-manufactured polycrystalline isotropic ice cores decrease (reversibly) with increasing temperature. All elastic properties and attenuation vary with ice temperature, but compressional-wave speed and attenuation prove most sensitive to temperature, indicative of pre-melting of the ice. This method of resonant ultrasound spectroscopy can be deployed in the field, for those situations where shipping samples is difficult (e.g. remote locations), or where the properties of ice change rapidly after extraction (e.g. in the case of sea ice).

## 1 Introduction

Ice sheets flow due to a combination of internal deformation and sliding at the base of the ice. The rate of internal deformation is strongly dependent on the englacial temperature, with flow rates increasing for warmer ice. The thermal regime in an ice body controls the onset of basal melting, a process which greatly increases basal sliding rates and therefore flow velocity (Hooke et al., 1980; Peters et al., 2012). Ice creep rate depends exponentially on temperature (Durham et al., 2010). An englacial temperature uncertainty of $5°$C corresponds to an uncertainty in internal deformation rates of a factor of two to five (using activation enthalpies for ice sheets Cuffey and Paterson (2010)). For frozen base scenarios (such as parts of Antarctica), the uncertainties on basal sliding rates that correspond to uncertainty on basal temperature will be of the same order of magnitude. Modelling techniques (Pattyn, 2010; Liefferinge and Pattyn, 2013) have been used to estimate the regional distribution of en-glacial temperature in large ice masses, but thermal profiles of ice sheets from bore holes are extremely limited, and come mainly from ice divides, with few observations from faster flowing ice (Peters et al., 2012). Englacial and basal temperatures across the vast majority of the Antarctic ice sheet are subject to uncertainties on the order of several degrees Celcius, limiting our ability to accurately model the contributions of internal deformation and basal sliding to ice sheet flow. Elsewhere,

geophysical methods (ice-penetrating radar and active-source seismology) can provide data on internal structure and physical properties of ice.

Seismic investigations of ice sheets (among others Bentley and Kohnen (1976); Horgan et al. (2011, 2008); Picotti et al. (2015)) present a potential window into the regional scale characteristics of ice bodies. Much focus has recently been placed on understanding the physical properties of ice that influence seismic wave propagation (Maurel et al., 2015). Of particular interest are the relationships of seismic wave attenuation (Peters et al., 2012; Gusmeroli et al., 2010, 2012) to the ice temperature.

Wave attenuation from tidal ($< 1\,\mathrm{Hz}$) to ultrasonic frequencies ($> 20\,\mathrm{kHz}$) in ice exhibits a strong sensitivity to temperature, particularly at high homologous temperatures close to the melting point (Matsushima et al., 2008; McCarthy and Cooper, 2016). In warmer glacial environments, such as mountain glaciers and the outlet ice streams of Western Antarctica, variation in attenuation (internal friction) is dominated by energy dissipation in grain boundary processes (Gribb and Cooper, 1998; Jackson et al., 2002; Kuroiwa and Yamaji, 1959; McCarthy et al., 2011; McCarthy and Cooper, 2016) and is thus strongly controlled by the density and the nature of grain boundaries, particularly grain boundary diffusivity. Ice can undergo pre-melting where water (or some modified form of water) exists on ice grain boundaries at temperatures potentially as low as $-30°\mathrm{C}$ (Hobbs, 1974). Antarctic ice-sheet thermal structures at ice divides (Engelhardt, 2004) show that the upper ice sheet is below pre-melt temperatures and the base is above pre-melt temperatures, imparting a strong mechanical contrast.

Laboratory measurements of the elastic and anelastic properties of materials can be used to calibrate and understand seismic field measurements (Watson and van Wijk, 2015). Here, we use resonant ultrasound spectroscopy (RUS) and time of flight ultrasound measurements to determine the dependence of the elastic and anelastic properties of polycrystalline ice on temperature.

The properties of elastic media can be represented by a stiffness tensor ($c_{ijkl}$) which relates the stress ($\sigma_{ij}$) applied to a sample with the resultant strain ($\epsilon_{kl}$):

$$\sigma_{ij} = c_{ijkl}\epsilon_{kl}, \tag{1}$$

which reduces to

$$\sigma_{\alpha} = c_{\alpha\beta}\epsilon_{\beta}, \tag{2}$$

when the Voigt recipe is applied (Watson and van Wijk, 2015). For elastically isotropic materials, the stiffness tensor can be reduced to two independent components and expressed as:

$$\begin{pmatrix} \sigma_{11} \\ \sigma_{22} \\ \sigma_{33} \\ \sigma_{23} \\ \sigma_{13} \\ \sigma_{12} \end{pmatrix} = \begin{pmatrix} \lambda+2\mu & \lambda & \lambda & 0 & 0 & 0 \\ \lambda & \lambda+2\mu & \lambda & 0 & 0 & 0 \\ \lambda & \lambda & \lambda+2\mu & 0 & 0 & 0 \\ 0 & 0 & 0 & \mu & 0 & 0 \\ 0 & 0 & 0 & 0 & \mu & 0 \\ 0 & 0 & 0 & 0 & 0 & \mu \end{pmatrix} \begin{pmatrix} \epsilon_{11} \\ \epsilon_{22} \\ \epsilon_{33} \\ 2\epsilon_{23} \\ 2\epsilon_{13} \\ 2\epsilon_{12} \end{pmatrix}, \tag{3}$$

where $\lambda$ and $\mu$ are the Lamé constants that define, together with density $\rho$, the isotropic P- and S-wave velocities as:

$$V_p = \sqrt{\frac{\lambda + 2\mu}{\rho}}, \qquad V_s = \sqrt{\frac{\mu}{\rho}}.$$ (4)

## 1.1 Forward modelling

The forward problem is to calculate the mechanical resonance frequencies of an elastic body for a given stiffness tensor, sample geometry, and density. In resonant experiments, sinusoidal excitation is applied to a sample at some point and its measured response is observed at some other point. Using the variational Rayleigh-Ritz method, we can calculate the displacement response of a sample to a sinusoidal point force applied at a particular location as a function of frequency (see Zadler et al. (2004) for a derivation of this relationship).

## 1.2 The inverse problem

The inverse problem is to estimate the elastic properties of a sample, given the measured resonant frequencies, dimensions, and density. An iterative Levenberg-Marquardt inversion method (Watson and van Wijk, 2015) adjusts the model parameters (the components of $c_{\alpha\beta}$) in order to minimize the difference between measured ($f^m$) and predicted ($f^p$) resonant frequencies in a least square sense. We calculate $\chi^2$ values to determine the goodness-of-fit of an isotropic model to our data by the relationship:

$$\chi^2 = \frac{1}{N} \sum_i^N w_i \left( \frac{f_i{}^m - f^p}{\sigma_i{}^d} \right)^2,$$ (5)

where $N$ is the number of measured modes, $\sigma^d$ is the estimated uncertainty in the repeatability of each measured mode, and $w_i$ is the weight given to each mode as a measure of the confidence (from 0 to 1) (Watson and van Wijk, 2015). We minimise $\chi^2$ for all our inversions to a narrow level of tolerance (12 - 14) to improve the comparability of our measurements. Starting values of $c_{\alpha\beta}$ for polycrystalline ice are taken from Gammon et al. (1983), and Gusmeroli et al. (2012).

## 1.3 Anelasticity

The quality factor $Q$ is a frequency-dependent measure of how rapidly wave energy is dissipated due to internal friction in the medium:

$$Q = 2\pi E/\Delta E = f_0/\delta f,$$ (6)

where $E$ is the energy in the wave field, $\Delta E$ is the energy lost per cycle due to dissipative mechanisms in the material, $f_0$ is the resonant frequency and $\delta f$ is the full peak width at half maximum amplitude (see box 5.7 in Aki and Richards, 2002). Because RUS operates in the frequency domain capturing all the internally scattered energy at the receiver (Watson and

van Wijk, 2015), $Q$ estimates derived from RUS are due to intrinsic attenuation alone. That is, intrinsic attenuation captures dissipative losses: energy that gets converted from elastic wave energy into heat or some other form.

## 2 Experimental setup

Ice was prepared using the "standard ice" method (Stern et al., 1997). Samples with a homogeneous foam texture (Fig. 1), a grain size of <1 cm, a random crystallographic preferred orientation (CPO), and nearly isotropic velocity characteristics were frozen in a cylindrical aluminium mould (70 mm internal diameter), and machined to 130 mm in length. The sample average density was 0.90 g/cm$^3$. We estimate the resulting ice samples had < 2 % porosity in pores of < 100 $\mu$m size. The microstructure of a sample of standard ice manufactured using the same method was characterised using electron backscatter diffraction (Prior et al., 2015). This method maps fully resolved crystal orientations and allows us to model the anisotropy of samples as it relates to crystal orientation. Characterising the entire sample microstructure, in this case, was not practical due to its size. We were, however, able to characterise a statistically significant number of grains (>4000) to make a robust prediction of anisotropy (using a Voigt-Reuss-Hill average), which we estimate at ≈0.1% for $Vp$. Additional EBSD analyses of samples made in this way (Prior et al. (2015), Vaughan et al. (2016, in prep), Qi et al. (2016, in prep)), all show a close to random CPO, with an average maximum $Vp$ anisotropy of <2%. The orientation of the maximum anisotropy in different samples is different, suggesting that the small anisotropy we see in EBSD data relates to a small sample volumes (of a few thousand grains). The whole sample used in this paper contains of the order of $1 \times 10^7$ grains, and small magnitude local effects will be averaged as isotropic in our columnar sample by the resonance method. In this study, the sample contains some micro-porosity with a non-homogeneous distribution. This may give rise to a small amount of anisotropy.

RUS experiments were performed in the setup depicted in Fig. 2, using a contact method outlined in Watson and van Wijk (2015). A function generator (Stanford Research Systems, DS345) sent a swept sinusoidal excitation (10 V peak to peak) to a contacting piezoelectric transducer (Olympus NDT 500-kHz V101/V151) centred on the sample's end. Coupling between the sample and transducers was ensured by a thin layer of low temperature silicon grease. The resulting oscillations propagate through the ice sample and were detected by another transducer centred on the opposite end of the sample. The transmitted signal was synchronously detected by a DSP Lock-in amplifier (Stanford Research Systems, SR850) and divided into an in-phase component and an out-of-phase component with the reference signal. The magnitude of the two components was recorded on a Tektronix oscilloscope (TDS 3014B) and transferred to a PC via an Ethernet connection.

The sample was mounted in a counter-balanced floating platform (Fig. 2b) to minimize load on the ice by the top transducer, as loading can influence mechanical resonance (Zadler et al., 2004). The apparatus and the sample were placed inside a chest freezer which was allowed to warm slowly (increasing linearly a ≈4°C per hour) from its minimum temperature. To determine sample temperature, an identical ice sample placed in the same part of the freezer was monitored by a two thermocouples frozen into its core. The temperatures were recorded on LabView software using a National Instruments cDAQ thermocouple module equipped with k-type thermocouples. We conducted RUS measurements on ice at temperatures between −26°C and −5°C ±0.5°C, sweeping from 5 to 65 kHz.

Travel-time measurements of elastic waves through the long central axis of a warming sample were performed with the same transducers, where an Olympus NDT pulser generated a 200V pulse with a central frequency of 500 kHz. An identical receiving transducer was connected to an oscilloscope to detect the transmitted wave-field, and was collected using a 32 wave-form stack.

## 3  Results

### 3.1  Travel-time measurements

Ultrasonic wave fields allow us to estimate the compressional wave speed $V_p$ as a function of ice temperature (Fig. 3a). The estimated arrival times in Fig. 3b result in $V_p = 3.80 \pm 0.01$ km/s at -25°C. Measurements at successively higher temperatures show that $V_p$ changes -2.2 m/s/°C. The arrival of the secondary (shear) wave is outside the displayed times, but was obscured by scattered compressional waves.

### 3.2  Resonance measurements

From the observed resonances of our ice core (Fig. 4a), we extract the first 10 resonant frequencies under 40 kHz to estimate the elastic constants as a function of temperature. The resonant frequencies and the associated amplitudes decrease monotonically with increasing ice temperature (Fig. 4b). Subsequent cooling restores the original resonant frequencies and amplitude of the resonance spectra, showing no signs of significant hysteresis. Repeat measurements at fixed temperature give resonant peak positions with a standard deviation of $\sigma^d = 70$ Hz (This estimate of $\sigma_d$ was derived from a limited number of repeated measurements, and is likely an optimistic representation).

For each ice temperature, we invert for the elastic characteristics by iteratively changing the elastic constants in order to reduce the misfit, scaled by the data uncertainty as defined in Eq. (5). The iterations were terminated for values of $\chi^2$ between 12 and 14 at each temperature (see Table 1 for the results at $T = -25°C$). We attempted to minimise the $\chi^2$ values for every inversion to a similar level in order to ensure the results for all temperatures would be comparable. In most cases, the fit of the inversions ceased to improve beyond a certain number of iterations. For some sets of observed frequencies, additional iterations may have resulted in better fits, but then theses inversions would not be comparable with those data sets that would not converge any further. The resulting range of $\chi^2$ values are those that represent a compromise between all the datasets. From this procedure, we estimate $c_{11} = 12.6 \pm 0.05$ GPa and $c_{44} = 3.6 \pm 0.04$ GPa for standard ice at $T = -25°C$.

The temperature dependence of the elastic properties is captured in Fig. 5a. Values of $c_{11}$ and $c_{44}$ decrease with increasing ice temperature. Estimates of $V_p$ from TOF and RUS in Fig. 5b indicate a difference in absolute value, while both decay monotonically with increasing ice temperature. $V_s$ and $c_{44}$, however, appear less sensitive to ice temperature than the compressional wave speed and $c_{11}$.

### 3.3 Anelasticity

A Matlab curve fitting algorithm (findpeaks.m, from MATLAB, 2016a) detects peaks in the recorded resonant spectra and the width of those peaks at half the maximum amplitude, providing the input to Eq. (6) to estimate the quality factor $Q$ of the ice. While $Q$ generally decreases with increasing temperature (Fig. 6), the temperature dependence of $Q$ for our ice sample presents a bimodal distribution in $Q$ values, and in the sensitivity of $Q$ to temperature. Resonances with overall higher $Q$ values appear more temperature dependent than resonant modes with an overall lower $Q$.

The Matlab based forward modelling code RUS.m (Fig, 2008) computes the modal shape (torsional, flexural or extensional), associated with each of our observed peaks. Modes with higher overall values of $Q$ – and higher sensitivity to temperature in the ice (show a greater spread in $Q$ with changing temperature) – are associated with extensional modes (Fig. 6). These modes are essentially an axial compression coupled to a radial expansion (Zadler et al., 2004). Torsional modes, on the other hand, generate rotations of the sample about the vertical axis, depending entirely on the sample's shear velocity. Flexural modes represent energy travelling along paths that are tilted with respect to the sample axis and generate compressional and shear displacements on the end of the sample by bending. We observe extensional modes to be less attenuating, but their attenuation is more temperature dependent than for modes dominated by shear motion (flexural, torsional).

## 4 Discussion

### 4.1 Frequency dependence of velocity and attenuation

It is difficult to derive a relationship for the frequency dependence of elastic wave velocity (dispersion) in ice from the literature by using published velocity data measured at different frequencies, as the materials from each experiment are different. Seismic measurements (Kohnen, 1974) represent estimates derived from bulk ice with temperature gradients and significant internal fabric variability. Ultrasonic velocity measurements come from natural samples with variable microstructure (Kohnen and Gow, 1979) or from synthetic bubble free ice with an unknown microstructure (Vogt et al., 2008). Seismic field studies of surface waves (Rayleigh and Love waves) sampling bulk ice with temperature gradients and significant internal fabric variability show strong dispersion at low frequencies ($< 100$ Hz) (Picotti et al., 2015). However, this type of dispersion results from the sampling of different depths with different frequencies. Long wavelengths sample the deeper (generally faster) ice. Increases in $Q$ with frequency is observed in the laboratory in low frequency experiments (McCarthy and Cooper, 2016) and in field experiments (Gusmeroli et al., 2010).

The method presented in this work is advantageous in that measurements are taken across a range of frequencies on the same sample, where its characteristics (to the limit of the manufacturing method) are controlled, and the microstructure has been characterised. Our estimates of $V_p$ from ultrasonic pulsed measurements ($10^6$ Hz) trend higher than the estimates from RUS at $10^5$ Hz (Fig. 5a), and we observe a general increase in $Q$ with increasing frequency for all modal types. Dispersion and attenuation are coupled by the Kramers-Kronig relations (O'Donnell et al., 1981). The observed increase in $Q$ and $V_p$ with frequency is consistent with a visco-elastic medium.

## 4.2 Temperature dependence and pre-melt

The vertical temperature profile of polar ice sheets is complex. While near surface temperatures are typically below $-20°$C, basal temperatures can approach the bulk melting point (Pattyn, 2010; Cuffey and Paterson, 2010; Engelhardt, 2004; Joughin et al., 2004; Iken et al., 1993). It follows that a significant temperature-induced flow viscosity gradient must exist within in large
ice masses, on top of other contributing factors such as crystalline fabrics, which induce mechanical anisotropy. As a result, the temperature dependence of the elastic properties of ice are of interest from static to ultrasonic frequencies, and have been explored extensively for understanding ice behaviour on icy satellites McCarthy et al. (2008); McCarthy and Castillo-Rogez (2013); McCarthy and Cooper (2016).

The observed temperature dependence in our travel-time estimates of $V_p$ are consistent with Vogt et al. (2008); Kohnen
(1974); Bentley (1972, 1971) and Bass et al. (1957). Our results indicate that, in the temperature range of interest, the compressional wave speed is more sensitive to temperature than the shear wave speed. Similarly, wave attenuation captured in the quality factor $Q$ exhibits greater temperature sensitivity in the extensional resonant modes.

The quality factor $Q$ for compressional wave dominated extensional modes is greater, and more sensitive to temperature changes, than for flexural and torsional modes associated with shear wave properties. It is well understood that porosity,
dislocation structures, the configuration of grain boundaries, and any crystallographic preferred orientation textures play an important role in the absolute value of visco-elastic dissipation (McCarthy and Castillo-Rogez, 2013; Cole et al., 1998) and elastic wave speeds (Maurel et al., 2015; Diez and Eisen, 2015; Gusmeroli et al., 2012) in ice.

Quasi-liquid films can form on ice grain boundaries at temperatures above -30°C (Dash et al., 1995) (although the exact temperature associated with the onset of pre-melting is subject to some uncertainty, influenced by impurities, grain boundaries
(McCarthy and Cooper, 2016), and the frequency of investigation), which leads to a dramatic increase in $Q^{-1}$, particularly above -20°C in pure ice (Kuroiwa, 1964). Here, we attribute pre-melt films developing at triple-junctions or on grain boundaries as the dominant mechanism for the changes in the values of the elastic properties and wave attenuation as a function of ice temperature. This is a more likely contributor than the dislocation damping mechanisms proposed to dominate at the highest temperatures (Cole et al., 1998; Cole, 1990; Cole and Durell, 1995, 2001; McCarthy and Cooper, 2016), since these samples
have not been subject to deformation. This has been observed previously by Spetzler and Anderson (1968) and Kuroiwa (1964) in laboratory resonant bar measurements, and in the field at seismic frequencies (Peters et al., 2012). The exact nature of grain boundary wetting in ice by a pre-melt 'fluid-like' phase is poorly characterised. Recent work exploring grain boundary complexions (see Cantwell et al. (2014) for a review) suggest that grain boundaries can undergo transitions (which include pre-melting at the highest temperatures, (Luo, 2008)) in interface properties such as mobility, structure, and cohesive strength.
These temperature dependent complexions could account for dramatic changes in the bulk characteristics of a poly-crystal.

## 5   Conclusions

Laboratory resonance measurements provide quantitative estimates of the temperature dependent elastic properties and wave attenuation of polycrystalline ice. Resonant ultrasound spectroscopy and travel-time measurements reveal wave dispersion

and attenuation, as well as the temperature dependence of these properties. The compressional wave speeds and its intrinsic attenuation are most sensitive to temperature, which we attribute to liquid phases on ice grain boundaries associated with pre-melting conditions. Applied to real ice cores, this approach can be used to calibrate sonic logging and seismic field data on ice sheets and glaciers. The RUS method can be deployed in the field, which is important in situations where shipping of ice

samples is difficult (e.g. remote locations) or where the properties of ice change rapidly after extraction (e.g., in the case of sea ice).

## 6 Data availability

The raw resonant ultrasound and travel-time data are freely available on-line through the Auckland University Physical Acoustics Lab website at:

http://www.physics.auckland.ac.nz/research/pal/wp-content/uploads/sites/13 /2016/05/RUS_data_files.zip

*Author contributions.* MV, DP and KVW conceived the experiments. MV and KVW conducted the experiments. MV and HB processed the data. MV created the figures and wrote the manscript with support from KVW, DP and HB

*Acknowledgements.* This research was supported by the Marsden Fund of the Royal Society of New Zealand (UOO1116) and a University of Otago Research Grant. MV was supported by a University of Otago postgraduate research scholarship. We thank Jim Woods, Peter Fleury,

Leo van Rens and Brent Pooley for ongoing engineering and technical support throughout this project.

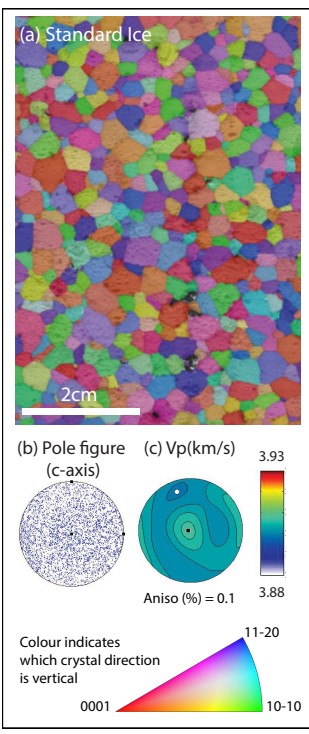

**Figure 1.** A subset of a large electron backscatter diffraction (EBSD) data set, from a sample of standard ice, manufactured by the same methods employed for the samples in these experiments. We acquired this map using a Zeiss Sigma VP FEGSEM fitted with on Oxford Instruments Nordlys camera and AZTEC software. Modifications required for cryo-EBSD are described in Prior et al. (2015).(a) Subsection of a large EBSD map of standard ice. The full map contains over 4000 grains. (b) C-axis pole figure in upper hemisphere projection, indicating the orientation of the c-axis at each pixel.(c) Vp model derived using a Voigt-Reuss-Hill average. The magnitude of anisotropy is indicated (0.1%)

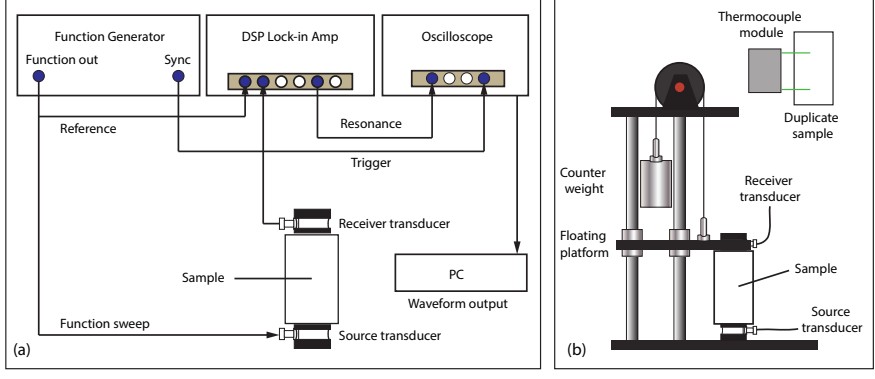

**Figure 2.** Diagram of the RUS setup (a) and of the load-minimizing sample frame with temperature monitoring equipment (b).

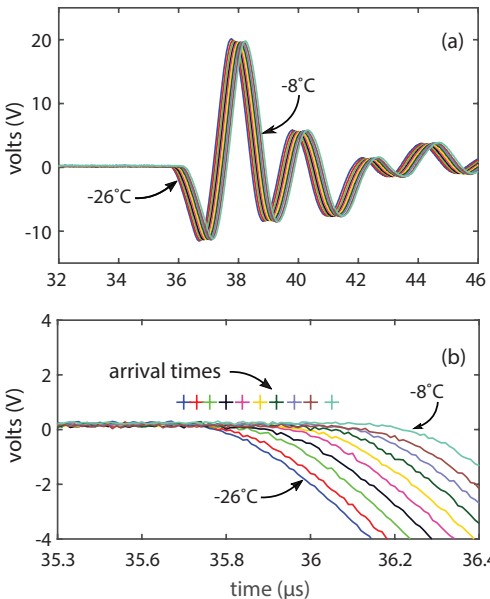

**Figure 3.** Ultrasonic waveforms (32 wave-form stack), transmitted through our ice cylinder, as a function of temperature (a), with a zoom of the first wave arrival in panel (b).

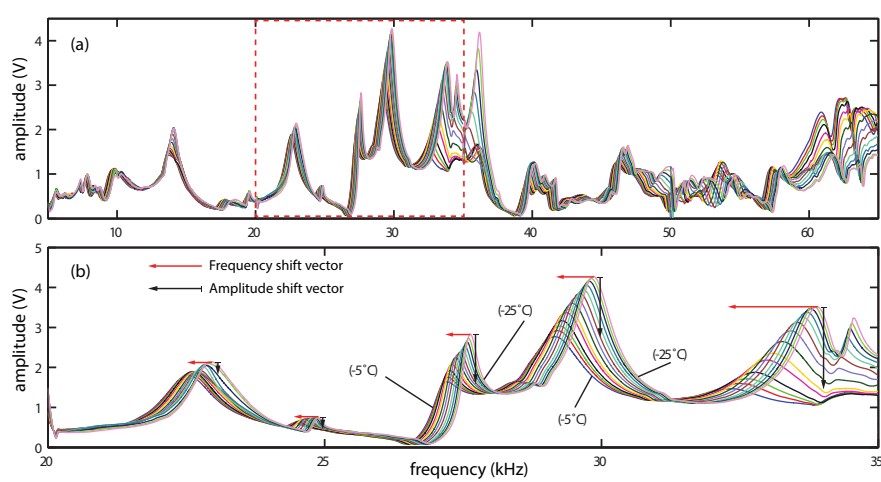

**Figure 4.** Resonant spectrum of our standard ice sample as a function of temperature (a). The range outlined by the red border is displayed in panel (b).

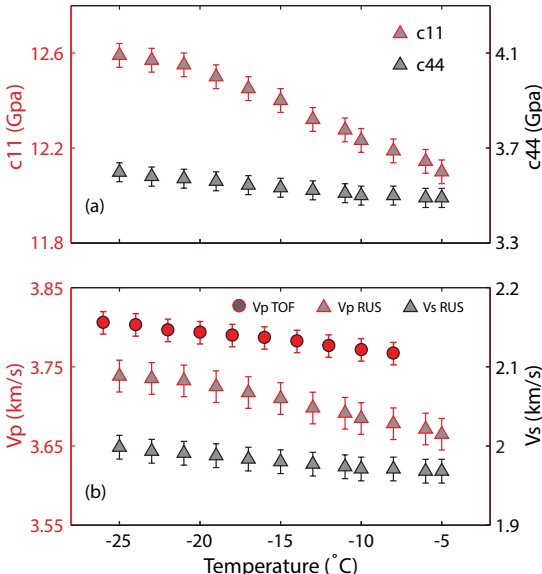

**Figure 5.** Estimates of the elastic constants $c_{11}$ and $c_{44}$ from RUS as a function of temperature (a). Estimates of $V_p$ and $V_s$ from RUS are compared to $V_p$ estimated from travel-time measurements in panel (b). The elastic constants are compared on the same vertical scale, as are Vs and Vp.

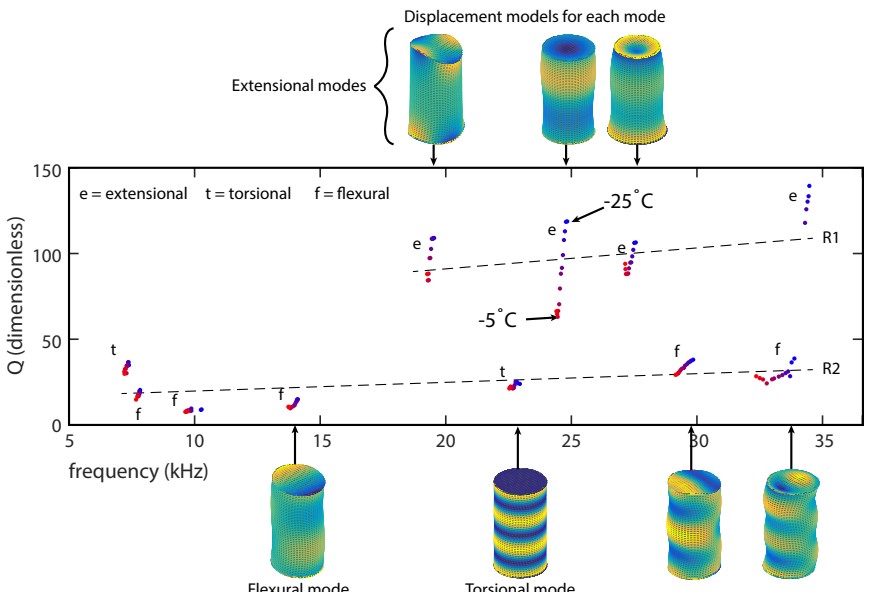

**Figure 6.** Quality factor $Q$ as a function of temperature and frequency for 11 resonant peaks. Each resonance mode is identified as flexural, extensional or torsional. R1 and R2 are linear regressions for the extensional modes and the flexural/torsional modes respectively, which show a general trend of increasing Q with $f$.

**Table 1.** Measured ($f^m$), initial-model predicted ($f_0^p$) and final-model predicted ($f^p$) resonant frequencies for our sample at -25°C. The final column is the relative contribution of each peak to the overall $\chi^2$.

| $f^m$(Hz) | $f_0^p$(Hz) | $f^p$(Hz) | $\left(\frac{f^m - f^p}{\sigma^d}\right)^2$ |
|---|---|---|---|
| 7386 | 7324 | 7334 | 0.55 |
| 7830 | 7334 | 7363 | 2.84 |
| 10256 | 11654 | 11629 | 76.94 |
| 14093 | 14794 | 14620 | 56.68 |
| 19583 | 19664 | 19580 | 0.00 |
| 22944 | 22328 | 22550 | 31.68 |
| 24871 | 25054 | 25008 | 3.83 |
| 27624 | 27893 | 27855 | 10.89 |
| 29850 | 29151 | 29729 | 2.98 |
| 33894 | 33800 | 33870 | 0.05 |

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
