# Peer review of "Monitoring the temperature dependent elastic and anelastic properties in isotropic polycrystalline ice using resonant ultrasound spectroscopy"

_The Cryosphere, 2016_

## Short Comment (SC1) · 31 May 2016

[revised manuscript text omitted]

---

## Referee Comment (RC1) · Anonymous Referee #1 · 6 Jul 2016

This paper presents measurements using resonant acoustic spectroscopy on an artificial ice core. The approach can be used to calibrate field data from seismic and sonic logging of ice sheets and glaciers. The experiment is carefully done, the paper is thoughtful and thoroughly referenced. The work is novel and in my opinion qualifies for publication, however I'd like to see the authors address a few points. My main complaint is that most of the conclusions are qualitative and it seems much more might have been done, even with the measurements already taken.

The authors estimate averaged elasticity parameters (disregarding mode conversion),

but it's really the temperature dependence, particularly for the anelastic properties, that is most useful for studies of real ice sheets. The authors explain that the Q values they found and the temperature dependence of those values exhibit bimodal distributions. I was expecting a plot of temperature dependence for the extensional modes at least, since these are relevant for seismic studies.

Why weren't measurements made below pre-melt temperatures, since as the text mentions this can cause an interesting mechanical transition? For pure ice this would be around -30 C, only 5 degrees colder than was measured (according to Peters et al. and references). Would it be possible to repeat measurements for other grain sizes or ice compositions, or speculate how grain size (curvature) and impurities affect premelting and Q? Perhaps this is planned in future work?

The paper explains that an identical ice sample in the same part of the freezer was monitored by thermocouples frozen into its core, as the temperature was slowly increased. How slowly? Is it possible that conduction through the thermocouple leads biased the temperature measurements? The authors might want to discuss this.

4.2 line 20 typo: "...must exist within in an ice body..."

---

## Author Comment (AC1) · 4 Aug 2016

**Response to comments: Monitoring the temperature dependent elastic and anelastic properties in isotropic polycrystalline ice using resonant ultrasound spectroscopy**

Matthew J. Vaughan[1], Kasper van Wijk[2], David J. Prior[1], and M. Hamish Bowman[1]

[1]Department of Geology, University of Otago, 360 Leith Walk, Dunedin, New Zealand, 9054.
[2]Department of Physics, Building 303, University of Auckland, 38 Princes Street, Auckland, New Zealand, 92019

*Correspondence to:* Matthew Vaughan (mattvaughan902@gmail.com)

**1 Introduction**

The following provides responses to all comments received during the on-line discussion of this manuscript, including those from reviewers and other short comments. We would like to thank all reviewers and members of the academic community for their contributions, and for providing valuable feedback during the open discussion period. Additional annotations from
5   attached supplements have been considered in the revised version of the manuscript.

**2 Response To SC1: Leighton Watson**

- Comment: "In line 8, page 3, it is stated that "iteration is terminated for values of chi-squared of between 12 and 14." Does this mean that the final value of chi-squared is between 12 and 14? These values seem large. Two possible explanations are (1) that the sample is not isotropic or (2) that the error estimates are overly optimistic. Do you have a sense about
10     which one of these explanations might be more important? Did you try to fit an anisotropic model e.g. VTI? And see if that improved the fit or increased the value of chi-squared (the latter is more likely)?"

Response: Yes, these are the final $\chi^2$ values. We attempted to minimise the $\chi^2$ values for every inversion to a similar level in order to ensure all the results would be comparable. In most cases, the fit of the inversions ceased to improve beyond a certain number of iterations. For some sets of observed frequencies, additional iterations of the inversions may have resulted in better
15   fits, but then theses inversions would not be comparable with those sets of peaks that would not converge any further. The resulting range of $\chi^2$ values are those that represent a compromise between all the datasets. We have made some changes to the text to reflect this in more detail (see track-changes file page 5, lines 15 to 25).

Given what we know about the microstructure of standard ice, it is most likely that our estimate of $\sigma_d$, which was derived from a limited number of repeated measurements (compared to the work of Watson and van Wijk (2015), for example) is a
20   lower bound.

We have expanded on our understanding of the (an)isotropy of standard ice from EBSD data, and our error estimates in the manuscript (see track-changes file page 4, lines 5 to 15). We have several EBSD analyses of samples made in this way (Prior et al. (2015), Vaughan et al. (2016, in prep), Qi et al. (2016, in prep), Seidemann, several unpublished data sets). All show a close to random CPO. When calculated (using a Voigt-Reuss-Hill approximation), average maximum $Vp$ anisotropy is <2%. The orientation of this maximum in different samples is different, suggesting that the small anisotropy we see relates to a small sample volume (of a few thousand grains). The whole sample used in this paper contains of the order of $1 \times 10^7$ grains and these small magnitude local effects will be averaged as isotropic in the columnar samples by the resonance method. Any uncertainty in anisotropy in the case of these samples, lies in the distribution of porosity, which, while limited, is almost certainly non-uniform in its distribution, and not aligned in any form of symmetry.

– Comment: "In addition, I am surprised by the relative contribution of the first peak to the overall chi-squared value, as shown in Table 1. It is very small. In the work by Kasper and I we found substantial misfit between the measured and predicted frequency of the first peak. This is described as a 'curious property' of RUS measurements by Migliori and Sarro (1997). I am intrigued why you do not see similar behavior."

Response: It is interesting that our first peaks gets a good fit, but this 'poor fit' phenomena for the lower frequency peaks has never been understood or explained, as made clear by the quote from Migliori and Sarro (1997). It may be possible the lowest peak is a resonance that is part of the system that includes the transducers/stand? In this case we are using samples much larger than those typically used in the literature or with rock, which may affect the influence the system on the lower frequency excitations.

– These comments also included a supplement in the form of an annotated .pdf, based on which, some additional edits to the manuscript were made. Of particular note was the following comment: "Any reasoning for attributing the dominant mechanism to pre-melt water films developing on grain boundaries?"

In response, we have included the following modified paragraph (see page 7, line 17 to 25 of the track-changes file):

– "Quasi-liquid films can form on ice grain boundaries at temperatures above -30°C (Dash et al., 1995) (although the exact temperature associated with the onset of pre-melting is subject to some uncertainty, influenced by impurities, grain boundaries (McCarthy and Cooper, 2016), and the frequency of investigation), which leads to a dramatic increase in $Q^{-1}$, particularly above -20°C in pure ice (Kuroiwa, 1964). Here, we attribute pre-melt films developing on grain boundaries as the dominant mechanism for the changes in the values of the elastic properties and wave attenuation as a function of ice temperature. This is a more likely contributor than the dislocation damping mechanisms proposed to dominate at the highest temperatures (Cole et al., 1998; Cole, 1990; Cole and Durell, 1995, 2001), since these samples have not been subject to deformation. This has been observed previously by Spetzler and Anderson (1968) and Kuroiwa (1964) in laboratory resonant bar measurements, and in the field at seismic frequencies (Peters et al., 2012)."

**3 Response To RC1: Anonymous Reviewer**

– Comment: "The authors estimate averaged elasticity parameters (disregarding mode conversion), but it's really the temperature dependence, particularly for the anelastic properties, that is most useful for studies of real ice sheets. The authors explain that the Q values they found and the temperature dependence of those values exhibit bimodal distributions. I was expecting a plot of temperature dependence for the extensional modes at least, since these are relevant for seismic studies."

Response: We would like to direct the reviewers attention to Figure 6 in the manuscript, where we have included a plot of $Q$ as a function of temperature and frequency for each mode, and identify whether the modes are extensional, torsional, or flexural excitations. Here, the bi-modal distribution in absolute values of $Q$ is clear, as is the greater sensitivity of extensional modes to changes in temperature.

– Comment: "Why weren't measurements made below pre-melt temperatures, since as the text mentions this can cause an interesting mechanical transition? For pure ice this would be around -30 C, only 5 degrees colder than was measured (according to Peters et al. and references). Would it be possible to repeat measurements for other grain sizes or ice compositions, or speculate how grain size (curvature) and impurities affect pre-melting and Q? Perhaps this is planned in future work?"

Response:

The temperature for the onset of pre-melting in ice is dependent upon several conditions, such as impurities, pressure, or even the mode of investigation, and has been proposed to occur at much warmer temperatures. In the classic work of Kuroiwa (1964), a dramatic increase in internal friction of pure ice is observed closer to -15°C in resonant bar experiments, while this change is observed at lower temperatures in natural samples and NaCl-doped ice. In our experiments, we see the most dramatic changes in both our elastic constants and values of $Q$ above -20°C. Arguably, the most drastic change in grain boundary mobility in deforming ice polycrystals occurs at -10°C (Duval et al., 1983), and is often associated with the proliferation of "a widened zone at grain boundaries with a liquid-like structure."

In any case, we were primarily limited by the minimum temperature of our freezer in this case (about -26°C).

– Comment: "The paper explains that an identical ice sample in the same part of the freezer was monitored by thermocouples frozen into its core, as the temperature was slowly increased. How slowly? Is it possible that conduction through the thermocouple leads biased the temperature measurements? The authors might want to discuss this."

Response: The temperature sweeps in these experiments took several hours to complete, and progressed approximately linearly at a rate of 4°C per hour. This has now been included in the text of the manuscript (see page 4, line 28 of the track-changes file). The thermal mass of the two thermocouples used to measure sample temperature is insignificant relative to the large ice sample, and should have very little influence on sample temperature.

– Comment: 4.2 line 20 typo: "...must exist within in an ice body..."

Response: This has been corrected. Thanks for pointing it out.

**References**

Cole, D., Johnson, R., and Durell, G.: Cyclic loading and creep response of aligned first-year sea ice, Journal of Geophysical Research: Oceans, 103, 21 751–21 758, 1998.

Cole, D. M.: Reversed direct-stress testing of ice: Initial experimental results and analysis, Cold regions science and technology, 18, 303–321, 1990.

Cole, D. M. and Durell, G. D.: The cyclic loading of saline ice, Philosophical Magazine A, 72, 209–229, 1995.

Cole, D. M. and Durell, G. D.: A dislocation-based analysis of strain history effects in ice, Philosophical Magazine A, 81, 1849–1872, 2001.

Dash, J., Fu, H., and Wettlaufer, J.: The premelting of ice and its environmental consequences, Reports on Progress in Physics, 58, 115, 1995.

Duval, P., Ashby, M. F., and Anderman, I.: Rate-controlling processes in the creep of polycrystalline ice, Journal of Physical Chemistry, 87, 4066–4074, <GotoISI>://A1983RM39800014, 1983.

Kuroiwa, D.: Internal Friction of Ice. I; The Internal Friction of H2O andD2O Ice, and the Influence of Chemical Impurities on Mechanical Damping, Contributions from the Institute of Low Temperature Science, 18, 1–37, 1964.

McCarthy, C. and Cooper, R. F.: Tidal dissipation in creeping ice and the thermal evolution of Europa, Earth and Planetary Science Letters, 443, 185–194, 2016.

Migliori, A. and Sarro, J. L.: Resonant ultrasound spectroscopy, John Wiley and Sons, 1997.

Peters, L. E., Anandakrishnan, S., Alley, R. B., and Voigt, D. E.: Seismic attenuation in glacial ice: A proxy for englacial temperature, Journal of Geophysical Research-Earth Surface, 117, doi:F02008 10.1029/2011jf002201, <GotoISI>://WOS:000302896500002, 2012.

Prior, D., Lilly, K., Seidemann, M., Vaughan, M., Becroft, L., Easingwood, R., Diebold, S., Obbard, R., Daghlian, C., Baker, I., et al.: Making EBSD on water ice routine, Journal of microscopy, 259, 237–256, 2015.

Spetzler, H. and Anderson, D. L.: The effect of temperature and partial melting on velocity and attenuation in a simple binary system, Journal of Geophysical Research, 73, 6051–6060, 1968.

Watson, L. and van Wijk, K.: Resonant ultrasound spectroscopy of horizontal transversely isotropic samples, Journal of Geophysical Research-Solid Earth, 120, 4887–4897, <GotoISI>://WOS:000359746700013, 2015.

---

## Referee Comment (RC2) · R. Cooper (Referee) · 24 Aug 2016

(Please note: I did my review reading the edited version of the manuscript posted by author Matthew Vaughan on 04 August 2016.)

This is a lovely piece of work; fun, really; and carefully explained. My compliments! There are a few very minor comments below, which the authors might wish to address at their discretion.

The significantly increased attenuation (decreased Q) noted exclusively for extensional (Youngs-modulus)-modes with increasing temperature, and its assignment to premelt-

ing at the grain boundaries is curious. I do not here argue against the idea – at least not directly; rather, I'll point-out behavior to which the current data might be compared.

We've done low-frequency (0.001 ≤ f (Hz) ≤ 1), high-temperature measurements of attenuation in silicate glass-ceramics and partial melts in both shear and (flexural) Youngs-modulus modes. In these materials, the quasi-equilibrium texture is to have melt (glass) confined to grain triple junctions (as a fully interconnected network, even at low volume fractions of melt/glass) and melt-free (i.e., crystalline) grain boundaries. As a consequence, shear attenuation was only modestly affected by the melt phase but Youngs-modulus attenuation was significant. In the latter, the pressure wave promotes/produces the relative motion of the melt and crystalline phases, following what the geodynamics community describes as compaction theory, as augmented with interfacial energy thermodynamics.

I'm no expert on premelting in ice. But experience with ice, consistent with what's seen in nature, is that water does not wet ice grain boundaries. A similar texture to silicate partial melts, rather, seems likely. Getting back to attenuation, if premelting in ice occurs at grain boundaries (or is uniquely associated with grain boundaries), would it not have two mechanical effects: (a) making degenerate the structures (and energetics) of the boundaries and so (b) affecting (presumably increasing) the shear attenuation more than the Youngs-modulus attenuation? Question is, what would be the thermodynamics and mechanics allowing premelting to be associated primarily with the extensional modes?

Perhaps this question is beyond the scope of the current manuscript. Nevertheless. . .

The recent work out of the materials community by scholars at Lehigh (Martin Harmer) and MIT (Craig Carter) on grain boundary 'complexions' is perhaps the key. If premelting does not represent a single quasi-liquid state but rather a host of states bridging a crystalline grain boundary at one extreme and a water film at the other, then the transformation(s) amongst states (and their kinetics), nicely described in the Harmer/Carter

efforts, could be an (the?) important aspect of extensional-mode attenuation.

Here are the little points (again, based on the 04 Aug 16 version):

Page 1-Line 4: It's Q that decreases with increasing temperature, not "attenuation" (which is 1/Q).

Page 3-Line 22: I do not understand "imperfections" in elasticity. The physics of elasticity do not change: you are measuring the stiffness of the bonds; it is by definition, perfect. Non-infinite Q means that the mechanical stimulus has sampled responses (anelastic +/- plastic) that dissipate, instead of store, strain energy.

Page 3-Line 25: I am not sure exactly what you mean by "intrinsic attenuation". "Intrinsic" usually means that which is solely a function of temperature (+/- pressure), i.e., independent of chemical potentials and texture. There are intrinsic effects—adiabatic loss; proton reordering in ice; point defect motion – but RUS can also "see" losses associated with texture, e.g., the presence of nonequilibrium defects like dislocations, grain boundaries, heterophase boundaries, etc. My experience with RUS has been to look at the impact of a finely disbursed second phase in synthetic peridotite, for example.

Respectfully,

Reid F. Cooper, Department of Earth, Environmental and Planetary Sciences, Brown University, Providence, Rhode Island, USA

---

## Author Comment (AC3) · 26 Sep 2016

**Response to comments: Monitoring the temperature dependent elastic and anelastic properties in isotropic polycrystalline ice using resonant ultrasound spectroscopy**

Matthew J. Vaughan[1], Kasper van Wijk[2], David J. Prior[1], and M. Hamish Bowman[1]

[1]Department of Geology, University of Otago, 360 Leith Walk, Dunedin, New Zealand, 9054.
[2]Department of Physics, Building 303, University of Auckland, 38 Princes Street, Auckland, New Zealand, 92019

*Correspondence to:* Matthew Vaughan (mattvaughan902@gmail.com)

**1 Introduction**

The following provides responses to all comments received during the on-line discussion of this manuscript from reviewer Dr. Reid Cooper (RC2) entitled 'The attenuation signature of pre-melting on grain boundaries in ice'.

**2 Response To RC2: Dr. Reid Cooper**

- **Comment:** 'The significantly increased attenuation (decreased Q) noted exclusively for extensional (Youngs-modulus)-modes with increasing temperature, and its assignment to pre-melting at the grain boundaries is curious. I do not here argue against the idea – at least not directly; rather, I'll point-out behaviour to which the current data might be compared. We've done low-frequency $(0.001 \leq (Hz) \leq 1)$, high-temperature measurements of attenuation in silicate glass-ceramics and partial melts in both shear and (flexural) Youngs-modulus modes. In these materials, the quasi-equilibrium texture is to have melt (glass) confined to grain triple junctions (as a fully interconnected network, even at low volume fractions of melt/glass) and melt-free (i.e., crystalline) grain boundaries. As a consequence, shear attenuation was only modestly affected by the melt phase but Youngs-modulus attenuation was significant. In the latter, the pressure wave promotes/produces the relative motion of the melt and crystalline phases, following what the geodynamics community describes as compaction theory, as augmented with interfacial energy thermodynamics.

  I'm no expert on premelting in ice. But experience with ice, consistent with what's seen in nature, is that water does not wet ice grain boundaries. A similar texture to silicate partial melts, rather, seems likely. Getting back to attenuation, if premelting in ice occurs at grain boundaries (or is uniquely associated with grain boundaries), would it not have two mechanical effects: (a) making degenerate the structures (and energetics) of the boundaries and so (b) affecting (presumably increasing) the shear attenuation more than the Youngs-modulus attenuation? Question is, what would be the thermodynamics and mechanics allowing premelting to be associated primarily with the extensional modes?

Perhaps this question is beyond the scope of the current manuscript. Nevertheless: The recent work out of the materials community by scholars at Lehigh (Martin Harmer) and MIT (Craig Carter) on grain boundary 'complexions' is perhaps the key. If premelting does not represent a single quasi-liquid state but rather a host of states bridging a crystalline grain boundary at one extreme and a water film at the other, then the transformation(s) amongst states (and their kinetics), nicely described in the Harmer/Carter efforts, could be an (the?) important aspect of extensional-mode attenuation.'

**Response:** Thank you, Dr. Cooper for your thoughtful and insightful comments, and for taking the time to consider our manuscript carefully.

The distribution of melt on grain boundaries and triple junction lines in ice remains unconstrained, and maybe data such as ours can be coupled with microstructural or atomic scale models to constrain this better. We must look elsewhere for an understanding of the nature and distribution of this 'fluid-like' phase. Your comments on complexions may provide insight here.

Recent work exploring grain boundary complexions (see Cantwell et al. (2014) for a review) suggest that in unary systems, grain boundaries can undergo first order, or continuous transitions which include prewetting / premelting at the highest temperatures (Luo, 2008), resulting in discontinuous changes in interface properties such as mobility, and cohesive strength. These temperature dependent transitions in the thickness, structure, and level of disorder of grain boundaries could account for dramatic changes in the bulk characteristics of a poly-crystal, and show that grain boundary mobility can increase dramatically even at temperatures in-sufficient to cause boundary wetting. One important point raised in experiments conducted by Schumacher et al. (2016) is that complexion transitions are also time dependent, with the rate of change of temperature during an experiment has influence on the stability of grain boundary complexions. The view of Luo (2008) was that the highest temperature grain boundary complexion is probably some kind of liquid-like melt film. In ceramics and metallurgy it is difficult to access this high temperature field, and ice might provide a great model material for work on this. In our experiments, we are able to achieve a homologous temperature of $> 0.95$. While we feel that the considerations of grain boundary complexions goes beyond the scope of this manuscript, we believe that the results obtained in our experiments raise important questions about the nature of grain boundary structures, and appreciate that this has been recognised by the reviewer. To this end, we have included some statements in the discussion directing readers to additional resources on complexions and the extensive work on low frequency measurements of attenuation in ice.

In the case of our experiments, we minimise the influence of dislocation related mechanisms, CPO, and second phases though a practised sample manufacturing process, performing 'zero' load experiments at frequencies betweens seismic and ultrasonic, using low amplitude oscillations that do not induce non-recoverable deformation. For these reasons, we localise the most likely contributor to drastic changes in attenuation to pre-melting effects.

We would like to point to a few key differences between our experiments, and lower frequency experiments involving creeping ice, such as in McCarthy and Cooper (2016) for example. In plastically deforming ice, particularly at higher homologous temperatures, creep can introduce an alignment of crystal orientations or a 'CPO' at low strain, and introduces attenuation mechanisms that depend on the generation, migration, and kinetics of defects in the lattice or grain boundaries. CPO in polycrystalline ice has been demonstrated to have a significant influence on $Q^{-1}$ (Cole et al., 1998). This influence arises because single crystals of ice can be anisotropic in their attenuation. Dislocation processes including dislocation damping, and grain boundary relaxation have been explored in the work of Cole and co-authors on saline and freshwater ice (Cole et al., 1998; Cole and Durell, 1995; Cole, 1990). This effort resulted in a temperature dependent model (Figure 1) that accounts for mul-

5 tiple mechanisms to explain attenuation behaviour. Importantly, the magnitude of $Q^{-1}$ in the dislocation damping regime in this model, likely depends on the amplitude of oscillations, and the location of the peak in grain-boundary dissipation likely depends on frequency of the oscillations.

[Figure]

**Figure 1.** Examples of attenuation in ice. (a) Schematic depiction of attenuation resulting from different mechanisms, after Cole and Durell (1995). (b) Compilation of field-based measurements of attenuation in ice sheets compared to laboratory experiments of Kuroiwa (1964) conducted on natural ice from Antarctica and Greenland. Frequencies differ among studies and range from 60 Hz to 1000 Hz. Note the relatively poor constraints on temperature for a given value of Q. References are B, Bentley (1971), BK, Brockamp and Kohnen (1965), BK76, Bentley and Kohnen (1976), CL, Clee et al. (1969), GS, Gusmeroli et al. (2010), JK, Jarvis and King (1993), K, Kohnen (1969), P, Peters et al. (2012), R, Robin and Robin (1958)

– **Comment:** Here are the little points (again, based on the 04 Aug 16 version):

Page 1-Line 4: It's Q that decreases with increasing temperature, not "attenuation" (which is 1/Q).

10 **Response:** This error has been corrected in the manuscript

– **Comment:** Page 3-Line 22: I do not understand "imperfections" in elasticity. The physics of elasticity do not change: you are measuring the stiffness of the bonds; it is by definition, perfect. Non-infinite Q means that the mechanical stimulus has sampled responses (anelastic +/- plastic) that dissipate, instead of store, strain energy.

**Response:** Thanks for pointing this out. We have edited this part of the manuscript to say 'dissipative mechanisms'

– **Comment:** Page 3-Line 25: I am not sure exactly what you mean by 'intrinsic attenuation'. 'Intrinsic' usually means that which is solely a function of temperature ($\pm$ pressure), i.e., independent of chemical potentials and texture. There are intrinsic effects: adiabatic loss; proton reordering in ice; point defect motion – but RUS can also "see" losses associated with texture, e.g., the presence of non-equilibrium defects like dislocations, grain boundaries, heterophase boundaries, etc. My experience with RUS has been to look at the impact of a finely disbursed second phase in synthetic peridotite, for example.

**Response:** What we mean by intrinsic attenuation, is absorption or dissipation. In RUS, we measure in the frequency domain over a period of time and all scattered energy arriving at the receiver is included in the calculations of Q, thereby isolating intrinsic Q from the attenuation due to scattering (Watson and van Wijk, 2015). In this context, we define intrinsic attenuation as dissipative losses: energy that gets converted from elastic wave energy into something else, usually associated heat. We have edited the manuscript to include the following:

'That is, intrinsic attenuation captures dissipative losses: energy that gets converted from elastic wave energy into heat or some other form.'

**References**

Bentley, C. R.: Seismic anisotropy in the West Antarctic Ice Sheet, Wiley Online Library, 1971.

Bentley, C. R. and Kohnen, H.: Seismic refraction measurements of internal-friction in Antarctic ice, Journal of Geophysical Research, 81, 1519–1526, doi:10.1029/JB081i008p01519, <GotoISI>://WOS:A1976BL02700008, 1976.

5    Brockamp, B. and Kohnen, H.: Ein Beitrag zu den seismischen Untersuchungen auf dem Grönlandischen Inlandeis, Polarforschung, 35, 2–12, 1965.

Cantwell, P. R., Tang, M., Dillon, S. J., Luo, J., Rohrer, G. S., and Harmer, M. P.: Grain boundary complexions, Acta Materialia, 62, 1–48, 2014.

Clee, T., Savage, J., and Neave, K.: Internal friction in ice near its melting point, Journal of Geophysical Research, 74, 973–980, 1969.

10   Cole, D., Johnson, R., and Durell, G.: Cyclic loading and creep response of aligned first-year sea ice, Journal of Geophysical Research: Oceans, 103, 21 751–21 758, 1998.

Cole, D. M.: Reversed direct-stress testing of ice: Initial experimental results and analysis, Cold regions science and technology, 18, 303–321, 1990.

Cole, D. M. and Durell, G. D.: The cyclic loading of saline ice, Philosophical Magazine A, 72, 209–229, 1995.

15   Gusmeroli, A., Clark, R. A., Murray, T., Booth, A. D., Kulessa, B., and Barrett, B. E.: Seismic wave attenuation in the uppermost glacier ice of Storglaciaren, Sweden, Journal of Glaciology, 56, 249–256, <GotoISI>://WOS:000280258200006, 2010.

Jarvis, E. and King, E.: The seismic wavefield recorded on an Antarctic ice shelf, Geophysical Press, 1993.

Kohnen, H.: Über die Absorption elastischer longitudinaler Wellen im Eis, Polarforschung, 39, 269–275, 1969.

Kuroiwa, D.: Internal Friction of Ice. I; The Internal Friction of H2O andD2O Ice, and the Influence of Chemical Impurities on Mechanical

20   Damping, Contributions from the Institute of Low Temperature Science, 18, 1–37, 1964.

Luo, J.: Liquid-like interface complexion: From activated sintering to grain boundary diagrams, Current Opinion in Solid State and Materials Science, 12, 81–88, 2008.

McCarthy, C. and Cooper, R. F.: Tidal dissipation in creeping ice and the thermal evolution of Europa, Earth and Planetary Science Letters, 443, 185–194, 2016.

25   Peters, L. E., Anandakrishnan, S., Alley, R. B., and Voigt, D. E.: Seismic attenuation in glacial ice: A proxy for englacial temperature, Journal of Geophysical Research-Earth Surface, 117, doi:F02008 10.1029/2011jf002201, <GotoISI>://WOS:000302896500002, 2012.

Robin, G. d. Q. and Robin, G.: Glaciology III: Seismic shooting and related investigations, Norsk Polarinstitutt, 1958.

Schumacher, O., Marvel, C. J., Kelly, M. N., Cantwell, P. R., Vinci, R. P., Rickman, J. M., Rohrer, G. S., and Harmer, M. P.: Complexion time-temperature-transformation (TTT) diagrams: Opportunities and challenges, Current Opinion in Solid State and Materials Science,

30   2016.

Watson, L. and van Wijk, K.: Resonant ultrasound spectroscopy of horizontal transversely isotropic samples, Journal of Geophysical Research-Solid Earth, 120, 4887–4897, <GotoISI>://WOS:000359746700013, 2015.

---

## Author Response (AR2)

**Response to final editor comments: Monitoring the temperature dependent elastic and anelastic properties in isotropic polycrystalline ice using resonant ultrasound spectroscopy**

Matthew J. Vaughan[1], Kasper van Wijk[2], David J. Prior[1], and M. Hamish Bowman[1]

[1]Department of Geology, University of Otago, 360 Leith Walk, Dunedin, New Zealand, 9054.
[2]Department of Physics, Building 303, University of Auckland, 38 Princes Street, Auckland, New Zealand, 92019

*Correspondence to:* Matthew Vaughan (mattvaughan902@gmail.com)

**1 Introduction**

**2 Response To Editor Comments: Prof. Eisen**

- **Comment:** 'I wonder if you find a more appropriate "second choice" term than Antarctica, as your results apply to basically every piece of ice.'

5

**Response:** Thank you, Prof. Eisen, for taking the time to make a final review of our manuscript. We have reworded the introduction slightly to be less specific, now including temperate mountains glaciers and the Greenland Ice Sheet.